# Impact of lower limb osteoarthritis on health-related quality of life: A cross-sectional study to estimate the expressed loss of utility in the Spanish population

Jesús Martín-Fernández[1,2,3]*, Roberto García -Maroto[4,5], Amaia Bilbao[3,6,7], Lidia García-Pérez[3,8], Blanca Gutiérrez-Teira[9], Antonio Molina-Siguero[10], Juan Carlos Arenaza[3,11], Vanesa Ramos-García[8], Gemma Rodríguez-Martínez[12], Fco Javier Sánchez-Jiménez[13], Gloria Ariza-Cardiel[1,3]

1 Unidad Docente Multiprofesional de Atención Familiar y Comunitaria Oeste, Gerencia Asistencial de Atención Primaria, Servicio Madrileño de Salud, Madrid, Spain, 2 Facultad de Ciencias de la Salud, Universidad Rey Juan Carlos, Madrid, Spain, 3 Red de Investigación en Servicios Sanitarios y Enfermedades Crónicas (REDISSEC), Spain, 4 Servicio de Cirugía Ortopédica y Traumatología, Hospital Universitario Clínico San Carlos, Servicio Madrileño de Salud, Madrid, Spain, 5 Doctorando en el Programa de Investigación en Ciencias Médico Quirúrgicas, Facultad de Medicina, Universidad Complutense de Madrid, Madrid, Spain, 6 Osakidetza, Hospital Universitario Basurto, Unidad de Investigación, Bilbao, Spain, 7 Instituto de Investigación en Servicios de Salud Kronikgune, Barakaldo, Spain, 8 Fundación Canaria Instituto de Investigación Sanitaria de Canarias (FIISC), Las Palmas de Gran Canaria, Spain, 9 Centro de Salud El Soto, Gerencia Asistencial de Atención Primaria, Servicio Madrileño de Salud, Madrid, Spain, 10 Centro de Salud Presentación Sabio, Gerencia Asistencial de Atención Primaria, Servicio Madrileño de Salud, Madrid, Spain, 11 Osakidetza, Hospital Universitario Basurto, Servicio de Traumatología y Cirugía Ortopédica, Bilbao, Spain, 12 Centro de Salud Infante Don Luis, Gerencia Asistencial de Atención Primaria, Servicio Madrileño de Salud, Madrid, Spain, 13 Centro de Salud Gregorio Marañón, Gerencia Asistencial de Atención Primaria, Servicio Madrileño de Salud, Madrid, Spain

* jmfernandez@salud.madrid.org

## Abstract

### Objective

Osteoarthritis of the lower limb (OALL) worsens health-related quality of life (HRQL), but this impact has not been quantified with standardized measures. We intend to evaluate the impact of OALL on HRQL through measures based on individual preferences in comparison to the general population.

### Methods

A cross-sectional study was designed. A total of 6234 subjects aged 50 years or older without OALL were selected from the Spanish general population (National Health Survey 2011–12). An opportunistic sample of patients aged 50 years or older diagnosed with hip (n = 331) or knee osteoarthritis (n = 393), using the American Rheumatism Association criteria, was recruited from six hospitals and 21 primary care centers in Vizcaya, Madrid and Tenerife between January and December 2015. HRQL was measured with the EQ-5D-5L, and the results were transformed into utility scores. Sociodemographic variables (age, sex, social group, cohabitation), number of chronic diseases, and body mass index were

**Data Availability Statement:** All relevant data are within the paper and its Supporting Information files.

**Funding:** This study has been funded by Instituto de Salud Carlos III through the Grants listed below, Co-funded by European Regional Development Fund, (ERDF) "A way of shaping Europe": JMF received the grant PI1300648, AB received the grant PI1300560, LGP received the grant PI1300518. The Foundation for Research and Biomedical Innovation of Primary Care of the Community of Madrid(FIIBAP, by its initials in Spanish) has subsidized the translation and publication costs of the manuscript.

**Competing interests:** The authors have declared that no competing interests exist.

considered. The clinical stage of OALL was collected using the Western Ontario and McMaster Universities Osteoarthritis Index and the Oxford hip score and Oxford knee score. Generalized linear models were constructed using the utility index as the dependent variable.

## Results

HRQL expressed by OALL patients was significantly worse than this of the general population. After adjustment for sociodemographic and clinical characteristics, the mean utility loss was -0.347 (95% CI: -0.390, -0.303) for osteoarthritis of the hip and -0.295 (95% CI: -0.336, -0.255) for osteoarthritis of the knee. OALL patients who were treated at a hospital had an additional utility loss of -0.112 (95% CI: -0.158, -0.065).

## Conclusion

OALL has a great impact on HRQL. People with OALL perceive a utility loss of approximately 0.3 points compared to the general population without osteoarthritis, which is very high in relation to the utility loss reported for other chronic diseases and for arthritis in general.

## Introduction

Health is a fundamental component of well-being in developed societies. The definition of health proposed by the World Health Organization in 1948 surpassed the biological concept of illness and included aspects of physical, psychological and social well-being [1], making it necessary to incorporate new measures to evaluate health status. Health-related quality of life (HRQL) appears to be a necessary measure for assessing an individual's perceived well-being. Although there is no universally accepted definition of HRQL, it has been suggested that measurements of HRQL should incorporate the subject's perception of his or her health situation, the impact of the disease or its treatment on his or her well-being and how that impact affects his or her functionality [2].

The assessment of HRQL is fundamental in evaluating the impact of disease and health intervention outcomes on both the individual and society as whole. Measures based on patient individual preferences are essential tools in these evaluations [3]. They allow patients to describe the impact of poor health and to calculate the "utility" scores (or rates) associated with each description of health status, making it possible to attribute value to such states through the study of preferences.

The most widely used tool for the measurement of HRQL in Spain is the EQ-5D [4], a questionnaire based on individual preferences that, through an algorithm, allows the attribution of a "utility" score to each described health state. This questionnaire is the most widely used tool for analyzing the cost-effectiveness of health care technologies [5]. Some European health technology assessment organizations, such as the National Institute for Health and Care Excellence (NICE) in the United Kingdom, have specifically stated that the EQ-5D is the preferred measure of HRQL in adults; thus, utilizing it ensures comparability among studies [6,7]. In Spain, the scores or utilities derived from the latest version of this tool, the EQ-5D-5L, have been proposed to provide information on economic evaluations of technologies [8].

The assessment of perceived health status in patients with chronic diseases is essential for measuring the impact and burden of disease, as in the case of patients with osteoarthritis of the lower limb (OALL) affecting the hip and knee. OALL represents a widely recognized public health burden. It is the eleventh most common cause of disability worldwide, and the disability-adjusted life years (DALYs) lost to it increased by 34.8% from 2005 to 2015 [9]. In Spain, a prevalence of 45.0% for knee osteoarthritis and 24.1% for hip osteoarthritis has been reported in people over 65 years of age [10]. Annual costs of €1,500 were estimated in 2007 for patients with osteoarthritis of the knee or hip in Spain, of which 86% were direct costs [11]. Other studies estimate costs of approximately €5,000 per year for patients in Europe and €12,000 per year in the US (2013 euros) [12]. Currently, the health care costs for generalized osteoarthritis account for between 0.25% and 0.50% of the Spanish gross domestic product (GDP) [13]. Furthermore, it is expected that the prevalence of OALL will continue to increase in the future, due largely to the aging of the population [14] and the increased prevalence of obesity [15]. As a result, this condition will likely continue to present substantial challenges for health planning in the coming years. Reliable measurements of health outcomes in OALL are increasingly important for health decision makers, health professionals and patients [16]. An essential element for advancing knowledge of the impact of OALL is the availability of measures that can be used to accurately and thoroughly measure the individual burden posed by this disease [17].

In Spain, HRQL has been assessed in patients with OALL who consulted trauma specialists [18] and in older people with OALL in the community setting [19], and the results suggest that this disease is associated with a marked decrease in perceived well-being; however, to date, no study has determined what differences exist between this group and the rest of the community to isolate the effects of the disease from those produced by concomitant sociodemographic or clinical conditions.

This study aims to evaluate the impact of OALL on HRQL through measures based on individual preferences in comparison to the general population that does not have osteoarthritis.

## Materials and methods

### Design

A cross-sectional study was designed. The included population was collected from a random sample of the Spanish general population and from a sample of patients with OALL. The general population data were obtained by selecting the population aged 50 years or older from the 2011–2012 National Health Survey (NHS), of which the methodological bases are publicly available [20]. Of the 21,007 available surveys, 6,234 responses of subjects 50 years or older who stated that they did not have a medical diagnosis of osteoarthritis, arthritis, or rheumatism were chosen. To study patients with OALL, information was obtained through an opportunistic sampling of patients diagnosed with osteoarthritis of the hip or knee according to the American Rheumatism Association criteria [21,22] who consulted traumatology and rheumatology specialists at six hospitals and 21 primary care centers in Vizcaya, Madrid and Tenerife between January and December 2015. Patients with malignant or other organic diseases or psychiatric disorders that hindered participation and those who could neither read nor understand Spanish were excluded. Of the total subjects included (n = 758), only those 50 years of age or older were selected (n = 724).

Data regarding people with OALL were collected directly from the patients, and data regarding the general population were extracted from the NHS.

This sample size allowed the construction of explanatory models that were appropriate for achieving the proposed objective [23].

## Dependent variable

To measure the perceived HRQL in both samples, the EQ-5D-5L was used [4]. In its most current version, the EQ-5D-5L [24] consists of two parts: a 0-to-100 scale to assess HRQL, the Visual Analog Scale (VAS) and a questionnaire comprising 5 questions or domains (mobility, self-care, carrying out usual activities, pain/discomfort, and anxiety/depression) with 5 response levels (ranging from 1, no problems, to 5, impossibility or severe problems). From these five questions, 3,125 health states are obtained, and the score associated with each state is the utility index. Initially, it was considered that the utility index should range between 0, a state equivalent to death, and 1, which represented perfect health. However, individual preference studies of different health states have identified states that are less preferable than death for the general population, which means this index can yield negative values (up to -0.421) using the algorithms published for Spain [8]. The reliability, validity and sensitivity of the EQ-5D-5L have been studied in patients with OALL in Spain [25].

## Independent variables

The age and sex of each subject and his or her social group, based on a six-category classification related to occupation (with group I being the highest and group VI the lowest) [26], were collected as sociodemographic variables. Information on cohabitation was included, and those who lived alone were differentiated from those living with someone (in any form of cohabitation).

Perceived health status was assessed using a Likert scale of five categories (very good, good, fair, poor and very poor) and the number of chronic diseases diagnosed. The patients' body mass index (BMI) and whether they had undergone a primary care or hospital consultation were also studied.

Only for the OALL group was it possible to determine whether there was unilateral or bilateral involvement and whether osteoarthritis had been diagnosed in the other large lower limb joint (hip or knee). The Western Ontario and McMaster Universities Osteoarthritis Index (WOMAC) [27] and the Oxford knee score (OKS) [28] and Oxford hip score (OHS) [29] questionnaires were used to characterize the clinical stage of OALL. The WOMAC is a multidimensional scale comprising 24 items measuring the domains of pain (five items), stiffness (two items) and physical function (17 items) in patients with OALL [27]. We used the version with five response levels, scored from 0 to 4, representing different degrees of intensity (none, mild, moderate, severe or extreme) for each item. Those scores are summed and standardized to yield a score from 0 to 100 (from better to worse capacity). The higher the score, the worse the patient's status is. This questionnaire has been adapted and validated in Spain [30]. The OKS and OHS scales measure the severity of symptoms in patients with osteoarthritis of the knee and hip, respectively. Each scale consists of 12 questions, and the scores, which range from 0 to 48 points, classify the clinical situation of osteoarthritis patients in 4 groups: excellent (> 41 points), good (between 34 and 41 points), moderate (between 27 and 34 points) and poor (<27 points) [31]. The Spanish versions of the OKS and OHS questionnaires have also been validated [32,33].

## Analysis

We present the descriptive statistics of the explanatory and dependent variables with their measures of central tendency and dispersion. The qualitative variables were compared using chi-squared tests, and the quantitative variables were compared using Student's t test or, if necessary, its nonparametric equivalent.

To address the main objective, generalized linear models (GLMs) were constructed, with the utility index attributable to the subject's perceived health status as the dependent variable. To select the best model, the Akaike information criterion (AIC) and the Bayesian information criterion (BIC) were examined. The Gaussian family and "identity" link function were selected as the more appropriate ones using these criteria. Standard errors (SEs) were calculated using robust methods to prevent possible heteroscedasticity [3,34].

Model 1 included variables related to the reported HRQL and the presence of OALL, model 2 added the affected joint, and model 3 also included whether the patient had been evaluated in the hospital setting. Improvements in model fit were compared by calculating Akaike weights, which express the probability that a new model is better than the set of tested models [35]. The improvement in the BIC was also studied according to the interpretations proposed by Kass and Raftery [36]. The variable "perceived health status" was not included as an explanatory variable due to the risk of overfitting when measuring the same construct as the dependent variable. Neither bilaterality nor having another joint group affected by OALL (hip or knee) improved the explanatory power of the final model (model 3). Patients with any missing data were not included in the models. The results section explains the number of subjects included in each model.

## Results

The results are presented for 6958 subjects aged 50 or older: 6234 from the general population without osteoarthritis and 724 (10.41% of the total) with a diagnosis of OALL. Of the patients diagnosed with OALL, 393 (54.28%) had a diagnosis of osteoarthritis of the knee, of whom 53 (13.49%) had a previous diagnosis of osteoarthritis of the hip. Of the 724 patients with OALL, 331 (45.72%) had a diagnosis of osteoarthritis of the hip, of whom 115 (34.74%) had a previous diagnosis of osteoarthritis of the knee. A total of 286 patients with OALL were recruited from primary care clinics (39.50%), and 438 (60.50%) were recruited from specialized care practices: 369 (50.97%) from traumatology and 69 (9.53%) from rheumatology practices.

Table 1 shows the characteristics of the total sample studied. Among the patients with OALL, women, people older than 65 years, and more disadvantaged social groups predominated. There was also a higher prevalence of overweight and obesity and of cohabitation with some type of partner.

The perceived health status and utility expressed by patients with OALL were significantly worse than those expressed by the general population in terms of the transformation of the EQ-5D-5L into utilities and the VAS (Table 1) and of each domain of the questionnaire (Table 2). The differences in HRQL were most pronounced for the domains mobility, carrying out usual activities, and pain/discomfort. The range of utilities for the expressed health states ranged from -0.416 (state 55555) to 1 (state 11111) for both the general population and the patients with OALL. In the case of patients with osteoarthritis of the knee, the worst health state reported was 44555, which has a utility value of -0.297. The responses to the VAS of the EQ-5D-5L ranged from 0 to 100 for both the general population and the patients with OALL (both hip and knee).

Table 3 shows the level of severity for patients with osteoarthritis of the knee or hip as measured with the WOMAC and the OHS/OKS [30,32,33]. Those with osteoarthritis of the knee more frequently had bilateral involvement, but those with osteoarthritis of the hip more commonly showed concomitant involvement of the other large lower limb joint. There were no significant differences between the patients with knee and hip osteoarthritis in the utilities or the level of severity measured with specific instruments. Only the VAS of the EQ-5D-5L

**Table 1. Characteristics of the studied sample.**

| | General population N = 6234 | Population with OALL N = 724 | p |
|---|---|---|---|
| **Age (%)** | | | |
| 50–54 years | 21.82 | 4.70 | <0.001 |
| 55–59 years | 18.53 | 8.15 | |
| 60–64 years | 15.77 | 13.12 | |
| 65–69 years | 13.97 | 16.57 | |
| 70–74 years | 9.62 | 17.54 | |
| 75–89 years | 8.87 | 20.99 | |
| 80–84 years | 6.26 | 13.40 | |
| ≥85 years | 5.17 | 5.52 | |
| **Age, mean (SD)** | 64.44 (10.81) | 70.93 (9.13) | <0.001 |
| **Sex (%)** | | | |
| Female | 47.06 | 62.71 | <0.001 |
| **Social group (%)** | | | |
| Group I | 11.64 | 7.12 | <0.001 |
| Group II | 7.93 | 4.56 | |
| Group III | 19.00 | 16.79 | |
| Group IV | 15.22 | 20.62 | |
| Group V | 32.02 | 29.38 | |
| Group VI | 14.20 | 21.53 | |
| **Cohabitation (%)** | | | |
| With partner | 61.93 | 66.99 | <0.001 |
| **Chronic diseases (%)** | | | |
| None | 17.85 | 0.00 | <0.001 |
| One | 22.44 | 55.11 | |
| Two | 20.82 | 30.39 | |
| Three or more | 38.88 | 14.50 | |
| **Chronic diseases, mean (SD)** | 2.34 (2.02) | 1.66 (0.91) | <0.001 |
| **BMI (%)** | | | |
| Underweight | 1.03 | 0.14 | <0.001 |
| Normal | 31.18 | 19.06 | |
| Overweight | 40.02 | 43.23 | |
| Obese | 27.77 | 37.57 | |
| **BMI, mean (SD)** | 26.69 (4.09) | 29.02 (4.72) | <0.001 |
| **Perceived health status (%)** | | | |
| Very good | 13.60 | 2.23 | <0.001 |
| Good | 55.25 | 22.67 | |
| Fair | 23.24 | 48.26 | |
| Poor | 6.42 | 21.56 | |
| Very poor | 1.49 | 5.29 | |
| **EQ-5D-5L, utilities, mean (SD)** | 0.924 (0.160) | 0.532 (0.287) | <0.001 |
| **EQ-5D-5L, VAS, mean (SD)** | 74.71 (17.71) | 56.31 (21.70) | <0.001 |

BMI: Body Mass Index; SD: Standard Deviation; VAS: Visual Analog Scale.

showed a trend of patients with arthritis of the hip presenting worse perceived HRQL than those diagnosed with osteoarthritis of the knee, although the difference was not significant.

Table 4 shows the results of the explanatory models for the expressed utilities. Model 3 emerged as the best model because its Akaike weights, compared to those of the set of models

**Table 2. Distribution of the responses to the different domains of the EQ-5-5L for the general population and the population with lower limb osteoarthritis (OALL).**

| EQ-5D-5L domain | Percentage of responses per level | | | | | p |
|---|---|---|---|---|---|---|
| **Mobility** | **1** | **2** | **3** | **4** | **5** | |
| General population n = 6234 | 84.62 | 7.92 | 4.23 | 2.29 | 0.93 | <0.001 |
| OALL n = 723 | 10.93 | 20.06 | 43.02 | 23.93 | 2.07 | |
| **Self-care** | **1** | **2** | **3** | **4** | **5** | |
| General population n = 6234 | 93.55 | 2.98 | 1.44 | 0.90 | 1.12 | <0.001 |
| OALL n = 722 | 28.39 | 27.70 | 31.58 | 11.36 | 0.97 | |
| **Usual activities** | **1** | **2** | **3** | **4** | **5** | |
| General population n = 6234 | 89.09 | 5.21 | 2.68 | 1.57 | 1.44 | <0.001 |
| OALL n = 722 | 16.48 | 25.21 | 36.70 | 16.62 | 4.99 | |
| **Pain/discomfort** | **1** | **2** | **3** | **4** | **5** | |
| General population n = 6231 | 76.46 | 14.35 | 6.55 | 2.33 | 0.32 | <0.001 |
| OALL n = 723 | 6.22 | 21.30 | 37.07 | 30.57 | 4.84 | |
| **Anxiety/depression** | **1** | **2** | **3** | **4** | **5** | |
| General population n = 6226 | 85.21 | 9.14 | 3.82 | 1.48 | 0.35 | <0.001 |
| OALL n = 716 | 44.13 | 23.18 | 18.30 | 11.31 | 3.07 | |

and model 2, had a value of 1; that is, there is statistical certainty that it is the best model. In terms of improvement in the BIC, the evidence against the goodness of fit of models 1 and 2 was very strong (with BIC differences well above 10 in both cases). The three models, especially model 3, considerably reduced the error variance, as shown by the value of McFadden's adjusted $R^2$.

**Table 3. Characteristics of patients with OALL included in the study.**

| | Patients with hip osteoarthritis n = 331 | Patients with knee osteoarthritis n = 393 | p |
|---|---|---|---|
| **Bilateral involvement (%)** | 28.40 | 42.24 | <0.001 |
| **Involvement of other large lower limb joint (hip or knee) (%)** | 34.74 | 13.49 | <0.001 |
| **State according to Oxford score (%)** | | | |
| Excellent | 4.56 | 3.32 | 0.348 |
| Good | 11.85 | 8.42% | |
| Moderate | 18.24 | 19.64 | |
| Poor | 65.35 | 68.32 | |
| **EQ-5D-5L utilities, mean (SD)** | 0.517 (0.303) | 0.543 (0.272) | 0.110 |
| **EQ-5D-5L VAS, mean (SD)** | 54.92 (21.78) | 57.47 (21.59) | 0.059 |
| **Oxford score, mean (SD)** | 22.72 (10.58) | 21.97 (9.94) | 0.163 |
| **WOMAC score, mean (SD)** | | | |
| WOMAC score, pain | 45.82 (22.63) | 47.09 (20.55) | 0.216 |
| WOMAC score, limitation | 52.65 (22.81) | 51.14 (20.99) | 0.178 |
| WOMAC score, stiffness | 48.33 (25.97) | 46.81 (25.29) | 0.213 |
| WOMAC score, total | 50.85 (22.02) | 49.95 (20.10) | 0.282 |

SD: Standard Deviation; VAS: Visual Analog Scale. WOMAC: Western Ontario and McMaster Universities Osteoarthritis Index.

**Table 4. Explanatory models of the differences in utility derived from the EQ-5D-5L.**

| | Model 1 | Model 2 | Model 3 |
|---|---|---|---|
| **Variable** | **Coef (CI 95%)** | **Coef (CI 95%)** | **Coef (CI 95%)** |
| **Level of care** | | | |
| Hospital care vs other | | | -0.112 (-0.158, -0.065) |
| **Lower limb OA** | | | |
| Hip vs none | | -0.406 (-0.444, -0.369) | -0.347 (-0.390, -0.303) |
| Knee vs none | | -0.365 (-0.396, -0.334) | -0.295 (-0.336, -0.255) |
| **Lower limb OA** | | | |
| Yes vs No | -0.384 (-0.409, -0.360) | - | - |
| **Age** | | | |
| 55–59 vs 50–54 years | -0.004 (-0.014, 0.006) | -0.004 (-0.014, 0.006) | -0.004 (-0.014, 0.006) |
| 60–64 vs 50–54 years | 0.007 (-0.003, 0.017) | 0.007 (-0.003, 0.017) | 0.007 (-0.003, 0.018) |
| 65–69 vs 50–54 years | 0.013 (0.002, 0.024) | 0.013 (0.002, 0.024) | 0.013 (0.002, 0.024) |
| 70–74 vs 50–54 years | 0.006 (-0.008, 0.020) | 0.005 (-0.009, 0.019) | 0.005 (-0.009, 0.019) |
| 75–89 vs 50–54 years | 0.004 (-0.012, 0.019) | 0.003 (-0.012, 0.018) | 0.003 (-0.012, 0.018) |
| 80–84 vs 50–54 years | -0.062 (-0.084, -0.040) | -0.062 (-0.084, -0.040) | -0.064 (-0.086, -0.042) |
| ≥85 vs 50–54 years | -0.148 (-0.182, -0.114) | -0.148 (-0.182, -0.114) | -0.150 (-0.184, -0.117) |
| **Sex** | | | |
| Male vs female | 0.021 (0.014, 0.029) | 0.022 (0.014, 0.030) | 0.022 (0.014, 0.030) |
| **Social class** | | | |
| Group II vs group I | -0.008 (-0.022, 0.006) | -0.009 (-0.022, 0.005) | -0.010 (-0.024, 0.004) |
| Group III vs group I | -0.021 (-0.034, -0.008) | -0.021 (-0.034, -0.008) | -0.021 (-0.034, -0.008) |
| Group IV vs group I | -0.017 (-0.029, -0.004) | -0.017 (-0.030, -0.004) | -0.017 (-0.030, -0.004) |
| Group V vs group I | -0.023 (-0.034, -0.012) | -0.023 (-0.035, -0.012) | -0.023 (-0.035, -0.012) |
| Group VI vs group I | -0.024 (-0.037, -0.010) | -0.024 (-0.038, -0.010) | -0.024 (-0.038, -0.010) |
| **Cohabitation** | | | |
| With partner vs alone | 0.010 (0.002, 0.018) | 0.010 (0.002, 0.019) | 0.010 (0.002, 0.019) |
| **Chronic diseases** | | | |
| One vs none | -0.012 (-0.020, -0.005) | -0.012 (-0.020, -0.004) | -0.010 (-0.018, -0.003) |
| Two vs none | -0.028 (-0.037, -0.019) | -0.028 (-0.037, -0.019) | -0.030 (-0.039, -0.021) |
| Three or more vs none | -0.094 (-0.103, -0.085) | -0.094 (-0.102, -0.085) | -0.093 (-0.102, -0.084) |
| **BMI** | | | |
| Underweight vs nonobese obesemal weight | -0.048 (-0.102, 0.006) | -0.047 (-0.101, 0.007) | -0.047 (-0.100, 0.006) |
| Overweight vs nonobese | 0.005 (-0.004, 0.013) | 0.004 (-0.004, 0.013) | 0.004 (-0.005, 0.012) |
| Obese vs nonobese | -0.028 (-0.039, -0.017) | -0.028 (-0.039, -0.018) | -0.029 (-0.040, -0.019) |
| **Characteristics of the model.**<br>**Family: Gaussian**<br>**Link function: Identity**<br>**Fami** | N = 6542<br>AIC = -5522.47<br>BIC = -5373.18<br>McFadden's $R^2_a$ = 0.549 | N = 6542<br>AIC = -5529.77<br>BIC = -5373.69<br>McFadden's $R^2_a$ = 0.553 | N = 6542<br>AIC = -5593.02<br>BIC = -5430.15<br>McFadden's $R^2_a$ = 0.562 |

CI: Confidence Interval; BMI: Body Mass Index.

Model 3 shows how the presence of hip osteoarthritis decreases utility by an average of 0.347 points for people who are similar in terms of age, sex, social group, state of cohabitation and burden of chronic illness. For patients with knee osteoarthritis, the decrease in this value is 0.295 points on average. For any type of osteoarthritis, being treated in the hospital environment is associated with an average decrease in utility of 0.112 points, adjusted for the characteristics mentioned above.

Factors associated with a lower level of utility, in addition to OALL and being treated in a hospital environment, included being older (over 80 years), belonging to a more disadvantaged social group, being female, being obese, living alone, and presenting more chronic conditions.

## Discussion

This is the first study in Spain to quantify the impact of OALL on HRQL through preference measures adjusted for variables that are known to be associated with changes in perceived HRQL, such as age, number of chronic diseases, sex and other social conditions, such as social group or potential loneliness. In addition, an algorithm to estimate the social rating or individual preferences for health states was used to measure this impact according to the latest version of the EQ-5D-5L [8], a tool with advantages over previous versions and whose Spanish version has shown excellent psychometric properties for patients with OALL [25]. OALL is associated with a substantial decrease in the utility attributed to the state of health when this association is adjusted for other, potentially confounding variables. Living with OALL implies a utility loss of approximately 0.30 on average compared to the general population without osteoarthritis, which exceeds the thresholds of clinical relevance obtained in Spain (0.07 points in the EQ-5D-5L utility index for subjects with nonsurgically treated OALL at the group level) [25]. On the other hand, as could be inferred, patients treated at the specialized level had an even worse perception of their HRQL, and their attributed utilities averaged 0.11 points lower.

Degenerative joint involvement was reported by the general population in the United Kingdom as one of the chronic conditions that most affects quality of life, behind only pain and anxiety/depression [37]. Similar data have been reported for the Canadian population [38]. The utility loss attributed to osteoarthritis in these studies has generally been approximately 0.10 points when adjusted for other variables that could affect perceived HRQL, a loss approximately three times higher than that produced by other chronic conditions, such as diabetes or asthma [38]. In Spain, when the EQ-5D-3L was used to evaluate HRQL, the study of osteoarthritis as a whole identified it as a chronic condition that decreases utility to the same extent as cardiac problems or diabetes mellitus, with a decrease in utility of up to 0.10 points compared to the general population over 65 years of age [39]. The literature has reported that OALL has significant impacts on utility, similar to those found in this study. In an Italian population, an average loss of utility of approximately 0.28 points was observed in patients with OALL compared to the general population [40]. In this case, utility was evaluated with the 3-level version of the EQ-5D, and adjustments were made only for age and sex. According to these authors, the impact of OALL on utilities was similar to that produced by osteoporosis with vertebral fracture or ankylosing spondylitis and was much higher than that produced by other chronic diseases that affect the musculoskeletal system, such as Sjogren's syndrome or systemic sclerosis [40]. These figures far surpass the sensitivity thresholds of the EQ-5D-5L obtained in Spain. The established minimal clinically important difference (MCID) at the group level in Spanish patients with nonsurgically treated OALL was 0.07 [25], and in other countries, this threshold was 0.09 for subjects with arthritis [38].

It has also been possible to study the association between the setting from which the patient is recruited and his or her health situation. The average utility score for patients treated in the hospital setting was 0.11 points lower. In Spain, HRQL has been assessed in OALL patients who consulted trauma specialists [18] and in those in the community [19]; however, the results were not comparable due to the use of different measurement tools. This is the first time that evidence has been presented regarding the different degrees of disease involvement among patients seen at different levels of care, and the results are consistent with the organization of the Spanish national health system, in which the first level of care serves as a gateway for services for different chronic diseases and for osteoarthritis in particular [41].

It is possible that the great impact of OALL on HRQL occurs because the disease affects several of the domains incorporated into the EQ-5D tool, such as pain and loss of function [42]. However, in this case, the data presented show the impairment of a psychological component captured in the anxiety/depression domain of the EQ-5D-5L. This impact of OALL on mental health has previously been described in the literature [43]. In addition, the mental health domain as a component of perceived HRQL has been shown to be associated with joint replacement in OALL [44], so it should be evaluated with special attention.

On the other hand, the results of this study do not allow us to say that patients with osteoarthritis of the hip have, in general, a worse perception of their state of health than those with osteoarthritis of the knee, as has been suggested in some studies [18]. Although this tendency could be suspected from the descriptive analysis of the data (the VAS of the EQ-5D-5L), it is not observed when the models are adjusted, and the confidence intervals of the coefficients of hip/knee osteoarthritis overlap considerably.

The remaining sociodemographic and clinical characteristics, which were adjusted to determine the association between OALL and the utilities of health states, behaved in ways that were previously known. Older people, those from more disadvantaged socioeconomic groups, those who live alone, women, obese people and those with more chronic conditions tended to have a worse perceived health status, as described in previous studies in Spain [45,46]. It has previously been shown that the association between osteoarthritis and reported HRQL is confounded by other variables, such as sociodemographic factors, chronic disease or obesity [19,37]; therefore, the results presented here can be considered a realistic approach to the study of the association between OALL and HRQL.

## Limitations

The design of this study has some limitations. Cross-sectional studies are problematic when establishing causal associations, although the effect of the main confounding factors collected in the literature has been assessed. Additionally, the sample of patients with OALL cannot be understood as representative of the population with this condition but was collected using opportunity criteria. We can affirm that sampling was carried out in diverse geographical sites and that the profile of the patients with OALL (predominantly female, older, and with a lower socioeconomic status and a higher prevalence of obesity) coincides with that reported in other Spanish [47] and European [19] studies. The fact that the participants may have had more severe disease than patients in the general population due to overrepresentation in the hospital setting was taken into account in the analysis. The general population from whom data were collected comprised people who did not report any type of osteoarthritis or arthritis diagnosis. The approach to the existence of chronic conditions could be considered a limitation of the study, as the conditions, or lack thereof, were reported by the subjects. However, even though there are problems with the general population data, the NHS constitutes the most representative and highest-quality information available at the time.

The EQ-5D-5L is a generic instrument designed to measure dimensions of health relevant to all health states, including healthy individuals, and not patients' perception of aspects of health specifically affected by OALL. However, it has the advantage of allowing us to attribute patients' preferences to the described health status in patients with OALL and to compare them between different populations [48].

## Implications

The importance of the results presented lies in the interest that health policy decision makers may have in reproducible and comparable quantifications of the impact of OALL on HRQL.

These results quantify the burden of the disease from the perspective of the patient. In addition, there are interventions, such as joint replacement surgeries, that involve a considerable investment of resources but can provide very valuable results from the perspective of the patient and society as a whole. It has frequently been noted that to assess the outcome of such interventions, it is necessary to use patient-reported outcome measures, such as quality of life, that help determine how to prioritize actions [49].

## Conclusion

OALL is a chronic disease that has a great impact on the HRQL of patients. Patients with OALL perceive a very significant loss of utility, approximately 0.3 points compared to the general population without osteoarthritis. This impact differs depending on the place where the patient has been treated. Patients treated in the hospital setting in a health system in which primary care functions as a gateway for health care services report an additional utility loss of 0.1 point. This utility loss attributable to OALL is very high in relation to what has been reported for other chronic diseases and for arthritis in general, exceeding the threshold of the so-called MCID by up to three times.

## Supporting information

**S1 Data. Data for Sharing.**
(XLS)

**S1 File. STROBE-checklist_cross-sectional.**
(DOC)

## Acknowledgments

We are grateful to colleagues in the participating hospitals and primary care centers for their support and to all patients for their collaboration.

## Author Contributions

**Conceptualization:** Jesús Martín-Fernández, Roberto García -Maroto, Amaia Bilbao, Lidia García-Pérez, Gloria Ariza-Cardiel.

**Data curation:** Jesús Martín-Fernández, Amaia Bilbao, Lidia García-Pérez, Blanca Gutiérrez-Teira, Antonio Molina-Siguero, Juan Carlos Arenaza, Fco Javier Sánchez-Jiménez, Gloria Ariza-Cardiel.

**Formal analysis:** Jesús Martín-Fernández, Amaia Bilbao.

**Funding acquisition:** Jesús Martín-Fernández, Amaia Bilbao, Lidia García-Pérez, Gloria Ariza-Cardiel.

**Investigation:** Jesús Martín-Fernández, Roberto García -Maroto, Amaia Bilbao, Lidia García-Pérez, Blanca Gutiérrez-Teira, Antonio Molina-Siguero, Juan Carlos Arenaza, Vanesa Ramos-García, Gemma Rodríguez-Martínez, Fco Javier Sánchez-Jiménez, Gloria Ariza-Cardiel.

**Methodology:** Jesús Martín-Fernández, Amaia Bilbao, Lidia García-Pérez, Vanesa Ramos-García, Gloria Ariza-Cardiel.

**Project administration:** Jesús Martín-Fernández, Amaia Bilbao, Lidia García-Pérez, Gemma Rodríguez-Martínez, Gloria Ariza-Cardiel.

**Visualization:** Jesús Martín-Fernández, Roberto García -Maroto.

**Writing – original draft:** Jesús Martín-Fernández, Amaia Bilbao.

**Writing – review & editing:** Jesús Martín-Fernández, Roberto García -Maroto, Amaia Bilbao, Lidia García-Pérez, Blanca Gutiérrez-Teira, Antonio Molina-Siguero, Juan Carlos Arenaza, Vanesa Ramos-García, Gemma Rodríguez-Martínez, Fco Javier Sánchez-Jiménez, Gloria Ariza-Cardiel.

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
