## [Decision Letter · Decision Letter 0]

4 Dec 2019

PONE-D-19-31220

Impact of lower limb osteoarthritis on health-related quality of life: An estimate of the loss of expressed utility in the Spanish population

PLOS ONE

Dear Dr. Martín-Fernández,

Thank you for submitting your manuscript to PLOS ONE. After careful consideration, we feel that it has merit but does not fully meet PLOS ONE’s publication criteria as it currently stands. Therefore, we invite you to submit a revised version of the manuscript that addresses the points raised during the review process.

We would appreciate receiving your revised manuscript by Jan 18 2020 11:59PM. To enhance the reproducibility of your results, we recommend that if applicable you deposit your laboratory protocols in protocols.io, where a protocol can be assigned its own identifier (DOI) such that it can be cited independently in the future. For instructions see: http://journals.plos.org/plosone/s/submission-guidelines#loc-laboratory-protocols

We look forward to receiving your revised manuscript.

Kind regards,

Yuanyuan Wang

Academic Editor

PLOS ONE

Journal Requirements:

Additional Editor Comments:

The reviewers have provided comments and suggestion which will help improve the manuscript.

Additionally, the authors need to provide more details about the inclusion and exclusion criteria for the OALL patients and participants without OALL from the general population. Are there any data about the accuracy or validation of the self-reported data on osteoarthritis/arthritis in the National Health Survey? It is not clear whether the data (as shown in the Table 1) were collected in the same way for the OALL patients and participants without OALL from the general population. It was stated that 393 patients had knee osteoarthritis and 331 had hip osteoarthritis. Does this means that no patients had both knee and hip osteoarthritis? If this is the case, what is the meaning of the data shown in the Table 3 for "Involvement of another large lower limb joint (hip or knee)"?

Reviewers' comments:

Reviewer's Responses to Questions

**Comments to the Author**

1. Is the manuscript technically sound, and do the data support the conclusions?

Reviewer #1: Yes

Reviewer #2: Partly

2. Has the statistical analysis been performed appropriately and rigorously? 

Reviewer #1: Yes

Reviewer #2: Yes

3. Have the authors made all data underlying the findings in their manuscript fully available?

Reviewer #1: Yes

Reviewer #2: Yes

4. Is the manuscript presented in an intelligible fashion and written in standard English?

Reviewer #1: Yes

Reviewer #2: Yes

5. Review Comments to the Author

Reviewer #1: Within their study, the authors report on the results of an observational study to assess the impact of having osteoarthritis of the lower-limb (OALL) on health-related quality of life, compared to the general population patients without OALL. Results are presented both without adjustment for patient characteristics, and with adjustment via a series of regression models. The manuscript is generally well-written, the data appears to be suitable, and it is good to see that the full dataset will be made publically available. I do have some concerns that the reporting of the regression models requires more information (and I’m not convinced about the choice of model 1).

Major comments.

• Data on those without OALL is from the 2011-2012 National Health Survey – did this include EQ-5D-5L responses? If so, explicitly state (I wasn’t aware that the EQ-5D-5L was available in 2011), if not, please state how these values were obtained (for example, if mapped from EQ-D-3L, the mapping algorithm used). If EQ-5D-5L values were indirectly obtained for the general population this should be noted as a limitation of the study.

• The author’s state that they have used generalised linear models, this is a very broad family of models! They provide details on the linear predictor used, but need to also describe the probability distribution used (Normal, Poisson, etc) and the link function (identity, log, etc).

• Given that the main objective of the paper is to estimate the impact of OALL on health-related quality of life compared to the general population, and Table 1 shows that there is a difference when not adjusting for patient characteristics, I was confused as to the purpose of Model 1, which does not include a variable for OALL. Shouldn’t Model 1 include OALL as a binary yes/no variable? Then Models 2 and 3 check if adding evidence on type of OALL and care setting improve fit.

• The authors provide evidence on the relative goodness of fit of the three models considered, but not on the models’ absolute goodness of fit. This could be achieved, for example, by visually comparing the observed EQ-5D-5L summary scores with the model-based estimates, potentially for the two sub-groups defined by the presence and absence of OALL. Further, a particular feature of EQ-5D-5L data is that there can be a ceiling effect (clusters at the summary value of one); are the models constrained to only predict values less than or equal to one? (As the primary objective is a comparison at a cohort level, having individual predictions outside the allowed range is a secondary problem, but still one worth noting if this occurs).

• Please complete the STROBE reporting guideline: https://www.equator-network.org/reporting-guidelines/strobe/

• Has consent been obtained releasing the patient-level data? If not, is it suitably anonymised?

Minor comments.

• It is unclear if a systematic search for existing utility measures for OALL has been performed? If so, this would strengthen the case for the originality of this work (some studies are discussed starting on line 116 of page 8, but it is unclear how they were identified).

• Table 1; for the continuous variables age, number of chronic diseases, and BMI, please also report the mean values per group.

• Table 3; please also add BIC values.

• For confidence intervals, please report as “xx to yy” instead of “xx - yy” as in the latter case this looks like a negative sign.

• Page 7, line 94: “…specifically request that the EQ-5D be used in all economic evaluations…” this is slightly overstating the point. NICE state a preference for EQ-5D, but acknowledge that there may be occasions when this is not the most appropriate measure, such as in children (see Sections 5.3.10 and 5.3.11 of the NICE reference case).

• Reference 9 is for the 2010 global burden of disease study – there are more recent versions of this study, which should have more recent evidence on the ranking of OALL on DALYs lost.

• Reference 8 has been subsumed into reference 7 in the bibliography.

Reviewer #2: The authors have done a great job in conceptualizing the study, outlining a clear aim and articulating the design and methods. Given that the EQ-5D-5l was used in quantifying HRQL, the manuscript will be greatly improved if the authors consider interpreting their findings based on MCID, as opposed to subjective phrases like "substantial" in describing differences in HRQL.

6. PLOS authors have the option to publish the peer review history of their article (what does this mean?). If published, this will include your full peer review and any attached files.

Reviewer #1: Yes: Benjamin Kearns

Reviewer #2: No

---

## [Author Response · Author response to Decision Letter 0]

23 Dec 2019

Dear editors:

We would like to thank you sincerely for the work invested by you and by the reviewers. The suggestions of the reviewers will be very useful for improving the quality of the revised manuscript. 

We have taken all comments into consideration and we have added a few general comments so that the methodology and the context can be better understood.

Editor Comments:

Additionally, the authors need to provide more details about the inclusion and exclusion criteria for the OALL patients and participants without OALL from the general population. 

We included an opportunistic sampling of patients aged 50 years or older, diagnosed with osteoarthritis of the hip or knee according to the American Rheumatism Association criteria who consulted traumatology and rheumatology specialists at six hospitals and 21 primary care centers in Vizcaya, Madrid and Tenerife between January and December 2015. We excluded patients with malignant or other organic diseases or psychiatric disorders that hindered participation, and those who could neither read nor understand Spanish. 

Exclusion criteria have been added to the manuscript.

The general population data were obtained by selecting the population aged 50 years or older from the 2011-2012 National Health Survey (NHS), who claimed not to have a medical diagnosis of osteoarthritis, arthritis, or rheumatism. The NHS was performed in 2,000 of the 35,960 sections of the 2011 Municipality Population Census, including 12 households per section area. Three-phase sampling, stratified by population size, was performed (census section, homes, and people in the home suitable for survey participation), to achieve a representative sample at a region level. These data are available everywhere and they can be checked at reference 20.

Are there any data about the accuracy or validation of the self-reported data on osteoarthritis/arthritis in the National Health Survey? It is not clear whether the data (as shown in the Table 1) were collected in the same way for the OALL patients and participants without OALL from the general population. 

We have no data about the validity of the information reported in National Health Survey, but the survey methodology poses the best approximation of the Spanish population’s health at present, and its information has been used in quite a lot of studies which approached to the health status of general population (Pinilla et al, PLOS One 2019; Lostao L et al PLOS One 2017; Martin-Fernández J et al, Gac Sanit 2017; Tamayo Fonseca et al BMC Health Serv Res 2017;...)

Data regarding to people with OALL were collected directly from the patients, and those referred to general population were extracted from the National Health Survey. This has been stated into Methods section.

It was stated that 393 patients had knee osteoarthritis and 331 had hip osteoarthritis. Does this means that no patients had both knee and hip osteoarthritis? If this is the case, what is the meaning of the data shown in the Table 3 for "Involvement of another large lower limb joint (hip or knee)"?

It was not clear enough in the manuscript, so we have remade the sentence:

“Of the patients diagnosed with OALL, 393 (54.28%) had a diagnosis of osteoarthritis of the knee, of whom 53 (13.49%) had a previous diagnosis of osteoarthritis of the hip. Of the 724 patients with OALL, 331 (45.72%) had a diagnosis of osteoarthritis of the hip, of whom 115 (34.74%) had a previous diagnosis of osteoarthritis of the knee.” 

Reviewer #1: 

Major comments.

• Data on those without OALL is from the 2011-2012 National Health Survey – did this include EQ-5D-5L responses? If so, explicitly state (I wasn’t aware that the EQ-5D-5L was available in 2011), if not, please state how these values were obtained (for example, if mapped from EQ-D-3L, the mapping algorithm used). If EQ-5D-5L values were indirectly obtained for the general population this should be noted as a limitation of the study.

The National Health Survey 2011-12 was developed between July 2011 and June 2012. It was the first National Survey in Spain which used the EQ-5D-5L for measuring the perceived health related quality of life (the latest National Health Survey 2017 did not use any standard tool for measuring HRQL). All the information about the NHS methodology can be found at:

https://www.mscbs.gob.es/en/estadEstudios/estadisticas/encuestaNacional/encuesta2011.htm

We got results from EQ-5D-5L for each subject included in the NHS and we used the algorithm proposed by Ramos-Goñi et al (Value in Health 2018) to calculate the utility values, both for general population and for OALL patients.

• The author’s state that they have used generalised linear models, this is a very broad family of models! They provide details on the linear predictor used, but need to also describe the probability distribution used (Normal, Poisson, etc) and the link function (identity, log, etc).

We agree with the reviewer. Several distributional families and link functions were tested and the one that best adjusted the dependent variable was chosen through the AIC and the BIC. The gaussian family and “Identity function” were selected as distributional family and link function respectively. This information has been added to Methods section.

• Given that the main objective of the paper is to estimate the impact of OALL on health-related quality of life compared to the general population, and Table 1 shows that there is a difference when not adjusting for patient characteristics, I was confused as to the purpose of Model 1, which does not include a variable for OALL. Shouldn’t Model 1 include OALL as a binary yes/no variable? Then Models 2 and 3 check if adding evidence on type of OALL and care setting improve fit.

We felt it is was a good suggestion, so we have changed table 4. Model 1 includes now OALL as a dichotomous variable. Coefficients are now different and “akaike weights” have been recalculated.

• The authors provide evidence on the relative goodness of fit of the three models considered, but not on the models’ absolute goodness of fit. This could be achieved, for example, by visually comparing the observed EQ-5D-5L summary scores with the model-based estimates, potentially for the two sub-groups defined by the presence and absence of OALL. Further, a particular feature of EQ-5D-5L data is that there can be a ceiling effect (clusters at the summary value of one); are the models constrained to only predict values less than or equal to one? (As the primary objective is a comparison at a cohort level, having individual predictions outside the allowed range is a secondary problem, but still one worth noting if this occurs).

Reviewer #1 suggested us to use techniques, which are more useful to assess the predictive ability of the models. But we assumed that we were building explicative models of the differences between groups. 

To test the global goodness of fit of each model, we have now estimated the McFadden's adjusted pseudoR squared. It mirrors the adjusted R-squared in OLS by penalizing a model for including too many predictors. 

The formula used was:

McFadden's adjusted R squared = 1 – [(Deviance model-k)/Deviance intercept], where k stands for the number of parameters (including the intercept). 

Mc Fadden’s adjusted R2 (or pseudo-R2) values were added for all models in table 4, so readers will be able to check the proportional reduction in “error variance”, as deviance plays a role analogous to the residual sum of squares in linear regression (Allison P at: https://statisticalhorizons.com/r2logistic). The interpretation of this Mc Fadden’s adjusted R2 has been added in Results section.

On the other hand, as Reviewer #1 mentioned, there is a current debate on how to manage the censored part of the utilities and on whether censoring regression methods are appropriate to make estimations over this point ( Sullivan PW. Med Decis Mak. 2011; 31(6):787–9). But subjects at perfect health state were not the target of this study. Besides, psychometric properties of the EQ-5D-5L in patients with hip or knee osteoarthritis in Spain were studied previously and minimal floor and ceiling effects were found (Bilbao A et al. Qual Life Res 2018; 27(11): 2897-2908.)

• Please complete the STROBE reporting guideline: https://www.equator-network.org/reporting-guidelines/strobe/

Strobe checklist has been added as “Supporting Information”.

• Has consent been obtained releasing the patient-level data? If not, is it suitably anonymised?

Data from OALL patients were collected after obtaining their written consent. Data from general population were obtained from a secondary anonimysed source, the National Health Survey. The National Health Survey microdata are publicly available at: https://www.mscbs.gob.es/estadEstudios/estadisticas/encuestaNacional/encuesta2011.htm

Minor comments.

• It is unclear if a systematic search for existing utility measures for OALL has been performed? If so, this would strengthen the case for the originality of this work (some studies are discussed starting on line 116 of page 8, but it is unclear how they were identified).

We did not make a systematic review for building the theoretical framework. We developed a search which aimed to map rapidly the key concepts underpinning the research area. We found primary sources which allowed us to develop secondary searches, so we are sure enough of the importance and novelty of this study. We could say that we performed a scoping review (Arksey, H & O'Malley, L, Journal of Social Research Methodology 2005; 8:1, 19-32).

• Table 1; for the continuous variables age, number of chronic diseases, and BMI, please also report the mean values per group.

Done

• Table 3; please also add BIC values.

BIC values have been added in table 4.

• For confidence intervals, please report as “xx to yy” instead of “xx - yy” as in the latter case this looks like a negative sign.

We have now used commas to separate the extremes of the confidence intervals. 

• Page 7, line 94: “…specifically request that the EQ-5D be used in all economic evaluations…” this is slightly overstating the point. NICE state a preference for EQ-5D, but acknowledge that there may be occasions when this is not the most appropriate measure, such as in children (see Sections 5.3.10 and 5.3.11 of the NICE reference case).

Reviewer #1 is right. The NICE recommendation was: ” Health effects should be expressed in QALYs. The EQ-5D is the preferred measure of health- related quality of life in adults.”

So we changed the original sentence to:

“[…]have specifically stated that the EQ-5D is the preferred measure of HRQL in adults; thus, utilizing it ensures comparability among studies [6,7].”

• Reference 9 is for the 2010 global burden of disease study – there are more recent versions of this study, which should have more recent evidence on the ranking of OALL on DALYs lost.

Thank you for the suggestion. We have changed the data by those provided by the Global Burden of Disease Study 2015 (Vos et al, Lancet 2016)

• Reference 8 has been subsumed into reference 7 in the bibliography.

It has been corrected

Reviewer #2: 

Overall, the study is well conceptualized with clearly articulated aim and methods. The statistical models employed are consistent with the outcomes of interest. Using the EQ-5D provides a unique advantage that the authors have not leveraged. Of particular note, the authors should consider discussing their findings in light of the clinical relevance based on Minimally Clinically Important Differences (MICD) that have been well documented for the EQ-5D versions. This is especially important for potential interpretation and policy implications of the study findings. 

The Minimal Clinical Important Clinically Important Differences (MICD) documented for the Spanish versión of the EQ-5D-5L was 0.07 points, although the ratios of the MCID and the MDC95% were over the unit, only at group level. So, only when we consider the change of a group we can understand that an improvement is significant if it is over 0,07 (Bilbao A et al, Qual Life Res 2018; 27(11): 2897-2908). We had already included this benchmark in Discussion section (page 20/ line 276 and page 21/ line 298) and we have now included a new reference in the Conclusion section.

Other minor, albeit important comments are listed below.

Materials & Methods:

1. Line 128 simply starts with the phrase: “Cross-Sectional Study”. If the intention of the authors is to indicate that the study design is a cross-sectional study, they should do so by making a complete statement. Please check the abstract for a similar.

The sentence has been drafted again in a complete way. Abstract section has been adjusted to 300 words length.

2. Line 162: The use of “Chronic diseases diagnosed” should be clarified. For instance, the authors may consider to list the number of chronic diseases included. Adjusting for concurrent chronic diseases is important in HRQL studies. However, even more important is the specific chronic conditions that are accounted for in the model.

We recognize that our approach to chronic disease was quite a bit weak. The multimorbidity phenomenon is characterized by its complexity. We knew that from an holistic perspective, the assessment of multimorbidity by using a simple list of diseases (weighted or not) may be not sufficient (Diederichs C et al, J Gerontol A Biol Sci Med Sci. 2011 Mar;66(3):301-11). But for feasibility reasons we only collected a list of chronic conditions and the same weight was attributed to each one.

Chronic conditions from patients with OALL were collected from their clinical records, based on the list of conditions from Charlson’s index. Information about chronic conditions from general population was collected asking for any chronic condition which was diagnosed by a physician prior to the interview. People were asked about a list which contained 30 chronic diseases and afterwards they were allowed to report any other chronic condition by means of an open question.

We think that this issue deserves a comment in the section refered to limitations, so we have added the next sentence: 

“The approach to the existence of chronic conditions could be considered a limitation of the study, as they were reported by the subject”.

3. Line 169 – 171, the author indicates that the WOMAC was used. Is it to be assumed that the English version was used? If a Spanish version was used, kindly provide more information about the version such as its construct validity and reliability (with references).

We used the WOMAC Spanish versión and its psychometric properties had been already referenced (line 183, reference 30).

Results:

1. Table 1: The multiplicity of p values < 0.0001 is an indication of a high likelihood of having type 1 error, potentially because of the large sample size. The author should consider accounting/adjusting for type 1 error in the study. 

We agree with Reviewer #2. If we were studying the similarities among groups it would be neccessary to adjust the type 1 error. But we knew that groups were not similar. We were looking for the characteristics of the subjects which should be included in the models in order to get the best adjust. So the p values are not considered as a whole, but each one is studied as a particular one.

2. The author has not defined Social Groups I to VI that are used in all tables. It is hard to tell which a higher social group is and which is lowest. 

Social group are defined on a six-category classification related to occupation (group I being the highest and group VI the lowest). This information has been added in Methods section.

3. Line 213: The use of older people (except for extreme ages) is not immediately obvious. The author should kindly indicate what age group(s) or ranges you are referring to. 

It has been clarified in Results section.

4. Table 4: 

- In the BMI category, the author should consider changing “normal” to “Non-obese”, given that underweight/overweight/obese people are also “normal”. 

The change has been done, we are sorry, we pretended to denominate this group as “normal weight”.

- The author included BMI and number of chronic diseases in their final model. Research indicates that multiple chronic diseases do cluster in obese populations. Was there a correlation between BMI and number of chronic diseases? If so, could there be an issue of collinearity?

In this case, the the Pearson correlation coefficient for the association between BMI and number of chronic conditions was 0.0929 (p<0.05), and it appears not to be relevant. The mean VIF uncentered values for the Models 1, 2 and 3 were 2.66, 2.58 and 3.78 respectively. So we thought we did not have any relevant collinearity problem in the showed models.

5. Line 251: The authors indicate that being treated in a “hospital environment is associated with an average decrease in utility..” What may be the reason for this findings? Is it possible that the reduced HRQL associated with hospitalization is because of less than quality care in the hospital environment? Or is it as a result of the fact that more severe cases of OALL are likely to be hospitalized? Kindly provide more details.

In the Spanish National Health System, family doctors usually acts as gatekeepers. So they attend patients with OALL in the first stages of the disease. Initial approach to OALL requires of changing lifestyles, avoiding increase weight and sendentarism, and the use of analgesia. This approach is usually achieved in primary care centres. When osteoarthritis progresses patient are referred to the specialized level. So it is not surprising that more severe cases were recluted in outpatient clinics, which depend on Hospitals (it does not imply that patient is hospitalized, unless a replacement surgery is indicated).

Discussion:

1. Line 254: The statement “This is the first study in Spain to quantify the utility loss attributed to OALL….” is not consistent with the study aim to evaluate the loss due to OALL (line 122). The authors should consider rephrasing the statement. 

Both sentences have been rephrased. So the new objective is: 

“This study aims to evaluate the impact of OALL on HRQL through measures based on preferences in comparison to the general population that does not have osteoarthritis.”

The objective has been also rephrased in the abstract section.

Discussion section starts with the next sentence: “This is the first study in Spain to quantify the impact of OALL on HRQL through preference measures, when adjusted for other variables that are known to be associated with changes in perceived HRQL…”

2. The statement “OALL is associated with a substantial decrease in the utility attributed to the state of health when this association is adjusted for other potentially confounding variables” is ambiguous at best. HRQL research using the EQ-5D usually qualifies quantitative decreases in the utility index as statistical or clinical differences based on a defined minimally clinical important difference (MCID). The authors should consider discussing their findings within the parameters of MCID. This has the potential for interpretation especially for clinical application. 

We agree with Reviewer #2. MCID had been mentioned in the first version of the manuscript, but we have now underlined the relationship between the loss of utilities in patients with OALL and the referred threshold in two paragraphs of the discussion and in the conclusión section.

3. The authors discussed “utility” as their outcome, contrary to the fact the HRQL was the outcome in the stated aim. Please consider re-phrasing related statements for clarity and consistency.

We understand “utility index” as an approach to HRQL assesed through preference based measures. Nevertheless we have reformulated the sentences as suggested by Reviewer #2.

4. Given the plethora of tests across each sub-group of the patient population and outcomes, the authors should consider adjusting for multiple testing or at least provide a rationale if they choose not to.

In table 1 repeated tests are used for describing groups. The aim of the study was not to describe groups but looking for explicative models which allowed us to understand the impact of OALL on utilities. As we discussed before the p-values should not be considered as a whole. They were only useful to understand why given individual characteristics will be included in the models, although we could omit the p values and doing the same development

We found that each p-value in table 2 (and in table 3 too) answer to a different particular question. In any case we did not mention the results of the p values and we did not find the need of adjusting for multiple tests as our main results and discussion will be based on models. 

5. The authors should discuss limitations of using the EQ-5D-5L and its potential implications on interpretation of the outcome (for instance a ceiling effect due to the upper limit of the score).

The limitation of using a generic instrument has been added in the appropriate section. 

On the other hand, as we discussed before, the EQ-5D-5L showed minimal floor and ceiling effects in Spanish patients suffering for OALL (Bilbao A et al. Qual Life Res 2018; 27(11): 2897-2908).

Implications:

Line 337: “These results provide quantification of the prevalence of the disease and the burden..” It is not clear what the authors refer to as the “prevalence of the disease”. The aim of the study was not to present the “prevalence” of the disease.

We agree with Reviewer #2. We have deleted the reference to prevalence as it was not our aim.

Besides the referred changes, the title was modified in order to explain the study design as recommenden by STROBE Statement. It has been underlined that patients with missing data were not included in the analysis, as it had been already explained in “Results” section the number of subjects included in each analysis.

Minor changes as footnotes have been added and marked, and a new edition review has been made.

We hope that these changes will contribute to improving the quality of the manuscript and the interest for potential readers.

On behalf of the research team,

Jesús Martín-Fernández

---

## [Decision Letter · Decision Letter 1]

15 Jan 2020

Impact of lower limb osteoarthritis on health-related quality of life: A cross-sectional study to estimate the expressed loss of utility in the Spanish population.

PONE-D-19-31220R1

Dear Dr. Martín-Fernández,

We are pleased to inform you that your manuscript has been judged scientifically suitable for publication and will be formally accepted for publication once it complies with all outstanding technical requirements.

With kind regards,

Yuanyuan Wang

Academic Editor

PLOS ONE

Additional Editor Comments (optional):

The authors have addressed all the comments properly.

Reviewers' comments:

Reviewer's Responses to Questions

**Comments to the Author**

1. If the authors have adequately addressed your comments raised in a previous round of review and you feel that this manuscript is now acceptable for publication, you may indicate that here to bypass the “Comments to the Author” section, enter your conflict of interest statement in the “Confidential to Editor” section, and submit your "Accept" recommendation.

Reviewer #1: All comments have been addressed

Reviewer #2: All comments have been addressed

2. Is the manuscript technically sound, and do the data support the conclusions?

Reviewer #1: Yes

Reviewer #2: Yes

3. Has the statistical analysis been performed appropriately and rigorously? 

Reviewer #1: Yes

Reviewer #2: Yes

4. Have the authors made all data underlying the findings in their manuscript fully available?

Reviewer #1: Yes

Reviewer #2: Yes

5. Is the manuscript presented in an intelligible fashion and written in standard English?

Reviewer #1: Yes

Reviewer #2: Yes

6. Review Comments to the Author

Reviewer #1: (No Response)

Reviewer #2: The authors have done an excellent job in responding to the comments. I would recommend Beefaroni's correction for multiple testing, given the number of tests performed per table for a total of four tables (20 tests in total). Thanks very much.

7. PLOS authors have the option to publish the peer review history of their article (what does this mean?). If published, this will include your full peer review and any attached files.

Reviewer #1: Yes: Benjamin Kearns

Reviewer #2: No

---

## [Editor Report · Acceptance letter]

17 Jan 2020

PONE-D-19-31220R1 

Impact of lower limb osteoarthritis on health-related quality of life: A cross-sectional study to estimate the expressed loss of utility in the Spanish population. 

Dear Dr. Martín-Fernández:

I am pleased to inform you that your manuscript has been deemed suitable for publication in PLOS ONE. Congratulations! Your manuscript is now with our production department. 

With kind regards,

on behalf of

Dr. Yuanyuan Wang 

Academic Editor

PLOS ONE